# Early administration of tecovirimat shortens the time to mpox clearance in a model of human infection

Bach Tran Nguyen[1]*, Aurélien Marc[1], Clara Suñer[2,3], Michael Marks[4,5,6], Maria Ubals[2,3,7], Águeda Hernández-Rodríguez[8,9], María Ángeles Melendez[10,11], The Movie Group¶, Dennis E. Hruby[12], Andrew T. Russo[12], France Mentré[1,13], Oriol Mitjà[2,3,14,15], Douglas W. Grosenbach[12], Jérémie Guedj[1]

1 Université Paris Cité, INSERM, IAME, F-75018, Paris, France, 2 Skin Neglected Diseases and Sexually Transmitted Infections Section, Hospital Universitari Germans Trias i Pujol, Badalona, Spain, 3 Fight Infectious Diseases Foundation, Badalona, Spain, 4 Clinical Research Department, London School of Hygiene & Tropical Medicine, London, United Kingdom, 5 Hospital for Tropical Diseases, London, United Kingdom, 6 Division of Infection and Immunity, University College London, London, United Kingdom, 7 Facultat de Medicina, Hospital Clinic, Universitat de Barcelona, Barcelona, Spain, 8 Microbiology Department, Clinical Laboratory North Metropolitan Area, University Hospital Germans Trias I Pujol, Badalona, Spain, 9 Department of Genetics and Microbiology, Autonomous University of Barcelona, Barcelona, Spain, 10 Microbiology Department, Hospital Universitario 12 de Octubre, Madrid, Spain, 11 Instituto de Investigación Sanitaria Hospital 12 de Octubre (imas12), Madrid, Spain, 12 SIGA Technologies, Inc., Corvallis, Oregon, United States of America, 13 Unité de Recherche Clinique, Hôpital Bichat, Assistance Publique-Hôpitaux de Paris, Paris, France, 14 Universitat de Vic-Universitat Central de Catalunya (UVIC-UCC), Vic, Spain, 15 School of Medicine and Health Sciences, University of Papua New Guinea, Port Moresby, Papua New Guinea

¶ Collaborator group listed in S1 Text.
* tran-bach.nguyen@inserm.fr

**Data Availability Statement:** All data are included in the Supplementary files.

**Funding:** BTN received funding from IAME UMR 137, Université Paris Cité, INSERM, F-75018, Paris,

## Abstract

Despite use of tecovirimat since the beginning of the 2022 outbreak, few data have been published on its antiviral effect in humans. We here predict tecovirimat efficacy using a unique set of data in nonhuman primates (NHPs) and humans. We analyzed tecovirimat antiviral activity on viral kinetics in NHP to characterize its concentration–effect relationship in vivo. Next, we used a pharmacological model developed in healthy volunteers to project its antiviral efficacy in humans. Finally, a viral dynamic model was applied to characterize mpox kinetics in skin lesions from 54 untreated patients, and we used this modeling framework to predict the impact of tecovirimat on viral clearance in skin lesions. At human-recommended doses, tecovirimat could inhibit viral replication from infected cells by more than 90% after 3 to 5 days of drug administration and achieved over 97% efficacy at drug steady state. With an estimated mpox within-host basic reproduction number, $R_0$, equal to 5.6, tecovirimat could therefore shorten the time to viral clearance if given before viral peak. We predicted that initiating treatment at symptom onset, which on average occurred 2 days before viral peak, could reduce the time to viral clearance by about 6 days. Immediate postexposure prophylaxis could not only reduce time to clearance but also lower peak viral load by more than 1.0 $\log_{10}$ copies/mL and shorten the duration of positive viral culture by about 7 to 10 days. These findings support the early

France (https://www.iame-research.center/). The funders had no role in study design, data collection and analysis, decision to publish, or preparation of the manuscript.

**Competing interests:** DWG, DEH, and ATR work for SIGA Technologies Inc. Other authors declare no conflicts of interests.

**Abbreviations:** IV, intravenous; NHP, nonhuman primate; PD, pharmacodynamics; PK, pharmacokinetics; SAEM, stochastic approximation expectation–maximization; TFMBA, 4-trifluoromethyl benzoic acid; VL, viral load; VPC, visual predictive check.

administration of tecovirimat against mpox infection, ideally starting from the infection day as a postexposure prophylaxis.

## Introduction

Declared in July 2022 a public health emergency of international concern, mpox (previously known as monkeypox) has been reported in more than 80,000 cases worldwide [1]. Classically, the virus is transmitted predominantly to human through direct contact with an infected animal, and less commonly between humans [2,3]. However, in the 2022 outbreak, mpox has been predominantly reported among men who have sex with men [1]. Little data are available on the natural history of mpox infection, and findings rely on scarce samplings from small cohorts [4–6]. In a study conducted in Spain, 77 individuals were enrolled shortly after symptom onset and followed until viral clearance [7]. Mpox was detected in various compartments, and viral replication was particularly high in skin lesions, with a median time to viral clearance of 25 days, and about 14 days to clear replication-competent virus.

Currently, tecovirimat is the most commonly used antiviral agent for mpox [3]. Initially developed to treat human smallpox, tecovirimat demonstrated strong antiviral activity by blocking the VP37 protein in virus maturation and release from the infected cell [8,9]. The drug pharmacokinetics (PK) and tolerance were assessed in animals and healthy volunteers to propose relevant dosing regimens [9,10], showing an oral bioavailability of 48%, higher with food intake, and high plasma protein binding (75% to 82%). Tecovirimat is metabolized mostly to 4-trifluoromethyl benzoic acid (TFMBA) with no pharmacologically active metabolites [11]. Based on these findings, tecovirimat was approved by FDA in 2018 and by Health Canada in 2021 for the treatment of smallpox. The drug was later approved in 2022 by European Medicines Agency for the treatment of smallpox, monkeypox, cowpox, and vaccinia complications [12]. However, only few data are available on its efficacy in mpox-infected patients [4,13]. Several randomized, placebo-controlled, double-blind trials are ongoing, but the assessment of the virological and clinical benefits of tecovirimat will be hampered by the ethical challenges in conducting randomized trials as well as the rapid reduction in the number of cases in most parts of the world.

Here, we analyze by modelling data collected in NHPs infected with mpox, in healthy volunteers treated with tecovirimat, as well as in untreated patients infected with mpox, to predict tecovirimat effects at different dosing regimen on mpox viral dynamics in human skin lesions and guide its current use.

## Results

### Modeling tecovirimat effect on mpox viral loads in NHPs

**Mpox viral kinetics.** After inoculation with $5 \times 10^7$ pfu of mpox Zaire 79 strain, all 24 macaques demonstrated high blood viral loads (VLs) (median VLs on D0: 6.5 [IQR 6.3 to 6.6] $\log_{10}$ copies/mL; data shown in S1 Fig). In this otherwise lethal mpox model, initiating tecovirimat on Day 4 postinfection (D4) led to rapid decline in VL and survival for all treated animals.

**Viral kinetic model and parameter estimates.** We used a target cell-limited model with an eclipse phase to fit observed viral dynamics in NHPs (Fig 1B, Materials and methods). The fitting process estimated the median within-host basic reproduction number $R_0 = 3.94$ (S1 Table). The productively infected cell death rate, $\delta$, was estimated at 0.3 per day. The estimate of $p$ depends on $T_0$, the initial target cell density, thus, only the product $p \times T_0$ could be

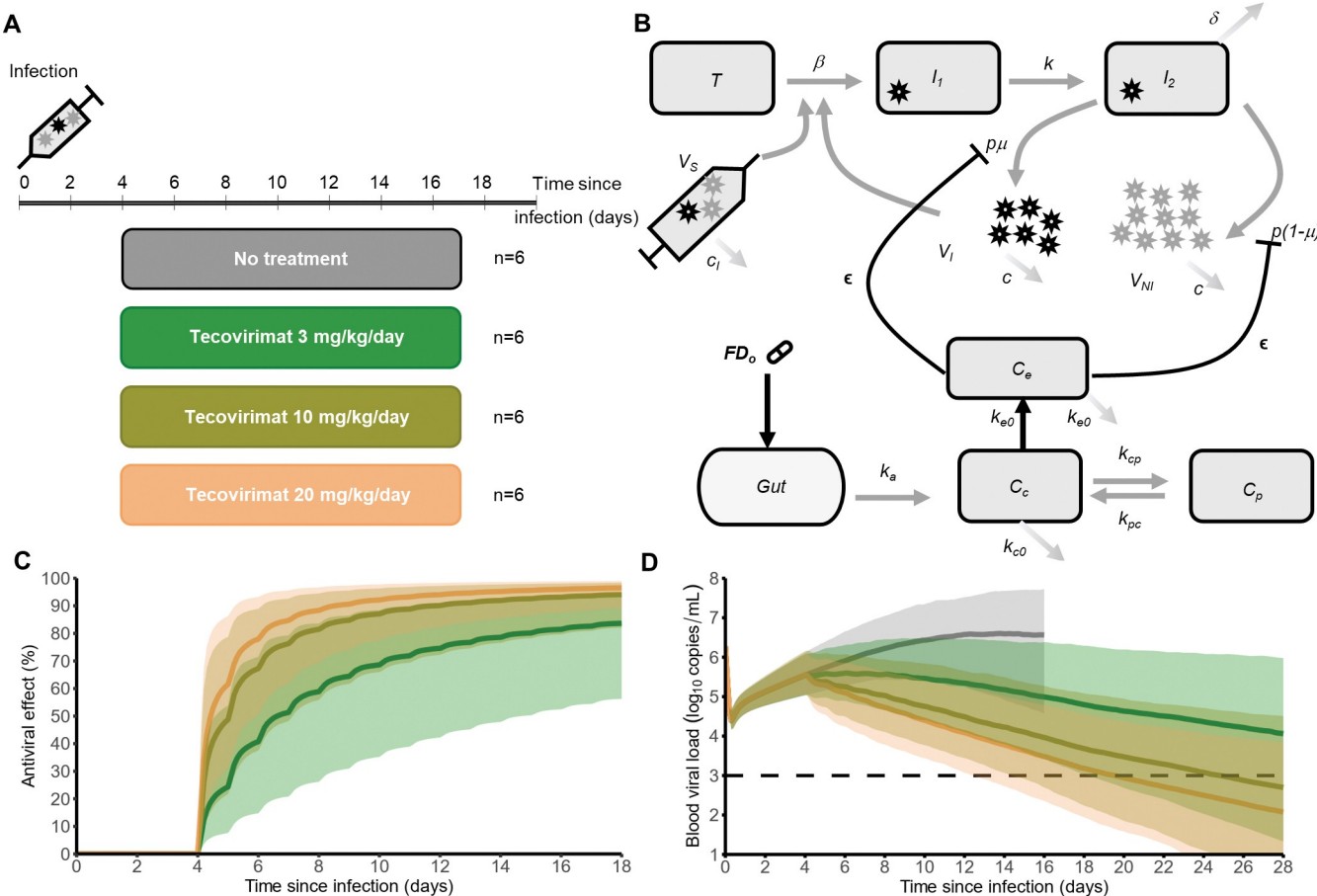

**Fig 1. Antiviral efficacy of tecovirimat in nonhuman primates (NHPs).** (**A**) Experimental study design. (**B**) Model structure: Target cells (T) are infected by infectious viruses, which can be produced de novo ($V_I$) or part of the inoculum ($V_s$) at rate $\beta$. Infected target cells enter an eclipse phase ($I_1$); after a median time $1/k$, after which they become productively infected cells ($I_2$) and produce $p$ viruses per day and are eliminated at rate $\delta$. Produced virus can be either infectious ($V_I$) with proportion $\mu$ or noninfectious ($V_{NI}$). Virions inoculated and produced de novo are cleared at different rates, noted rate $c_I$ and $c$, respectively. Tecovirimat is administered orally, absorbed at a first-order rate $k_a$ with a bioavailability $F$, and distributed in central ($C_c$) and peripheral ($C_p$) compartments. Drug transfer between 2 compartments is represented by constants $k_{cp}$ and $k_{pc}$ and elimination from the central compartment by constant $k_{c0}$. An effect compartment ($C_e$) represents the pharmacological effect site where tecovirimat inhibits viral production $p$ from infected cells at an efficacy $\epsilon$. (**C**) Tecovirimat predicted efficacy over time. (**D**) Mpox plasma viral load predicted by the model. In (**C**) and (**D**), predictions are given as median and 90% prediction interval. Grey: no treatment; green: 3 mg/kg/day; olive: 10 mg/kg/day; orange: 20 mg/kg/day; dashed horizontal line: limit of quantification.

reliably estimated ($2.5 \times 10^8$ copies/mL/day), with sensitivity analysis being performed (S2 Fig).

**Estimating tecovirimat antiviral effect.** We reconstructed tecovirimat concentrations in NHP plasma, $C_p(t)$, using a previously developed model [14]. The model accurately reproduced dose-depending plasma concentrations (S3 Fig). Next, in this model, the first-order rate constant between plasma and the effect compartment, noted $k_{e0}$ (estimated at 0.004 per day), describes the temporal dissociation of the time course between plasma concentrations and its inhibitory effect. Accordingly, we estimated tecovirimat $EC_{50}$, to 1.6 ng/mL, with sensitivity analyses being performed with the coefficient $h$ fixed at different values (S4 Fig). Alternatively, the concentration required to achieve the 90% antiviral effect, $EC_{90}$, was equal to 14.4 ng/mL. Although tecovirimat rapidly achieved high concentrations in plasma, an effect compartment was needed to capture the time to achieve high efficacy at the effect site. Thus, at doses of 10 and 20 mg/kg/day, model predicted that a median of 8 and 4 days, respectively, is required to

reduce viral production by more than 90% (Fig 1C). At steady state, the model predicted median antiviral efficacy of 84%, 94%, and 97% at doses of 3, 10, and 20 mg/kg/day, respectively. Accordingly, tecovirimat administration led to a rapid viral decline in all treated animals, with a median time to viral clearance of 25 and 20 days at doses of 10 and 20 mg/kg, respectively (Figs 1D and S1).

## Tecovirimat PK/PD in humans

Assuming the same PK/PD relationship holds in humans, we next estimated the antiviral efficacy that can be obtained in humans. For that purpose, we relied on a PK model developed in healthy volunteers (S2 Table) to predict drug exposure in both plasma (Fig 2A) and in the effect compartment at doses of 400 mg bid, 600 mg bid, or 600 mg tid. Concentrations in the effect compartment, $C_e(t)$, rise gradually to values ranging from 40 to 100 ng/mL depending on the dose administered, i.e., much higher than the estimated drug $EC_{90}$, leading to similar antiviral efficacy for all doses (Fig 2B). Accordingly, the antiviral activity took 3 to 5 days to reach 90% efficacy and, at steady state, achieved median levels of 98% for all dosing group. These predictions remained similar when considering large range of weights and recommended weight-adjusted doses (S5 Fig).

**Modeling mpox viral dynamics in human skin lesions.** Next, we characterized mpox kinetics in 54 non-antivirally treated individuals [7] for which the (putative) infection date had been identified (S3 Table). We focused on skin lesions, with a total of 264 samples included in our analysis. The delay between infection and symptom onset ranged from 0 to 14 days, with a median of 6 days (IQR = 4 to 8) (Fig 3B). The time from symptom onset to first PCR varied between 3 and 66 days, with a median of 23 days (IQR = 12 to 34). Patients were all male and aged 23 to 88 years, with a median of 35.5 years (IQR = 31 to 45.5).

**Viral dynamic modeling.** A target cell-limited model with an eclipse phase described well VL in skin lesions (Fig 3A). Viral dynamic parameters and their variability were estimated with good precision (Table 1). The visual predictive check (VPC) showed a central trend and predicted variability consistent with observed data (S6 Fig). The loss rate of productively infected cells, $\delta$, was equal to 0.64/day, corresponding to a half-life of infected cells of 25 hours (IQR = 22 to 30) in all patients. Similar to NHPs, with $\mu$ and $T_0$ fixed to frequently reported

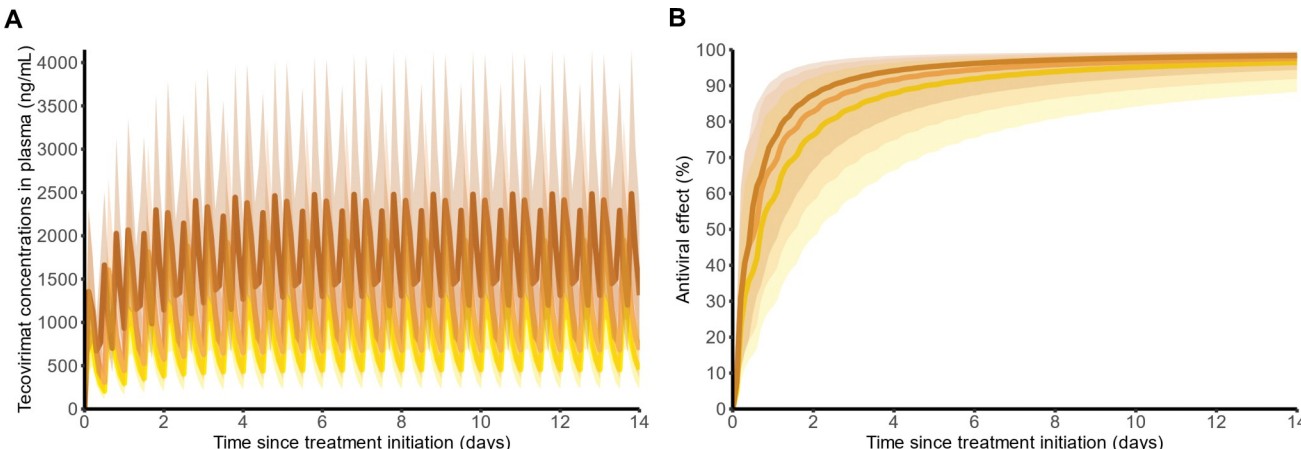

**Fig 2. Projecting tecovirimat pharmacokinetic and pharmacodynamics in mpox-infected humans.** (**A**) Tecovirimat plasma concentrations over time for different dosing regimen. (**B**) Tecovirimat antiviral effect over time. Predictions shown as median and 90% prediction interval, assuming a 14-day treatment course for a 78.4-kg patient, using pharmacokinetic parameters estimated in healthy volunteers (S2 Table), and pharmacodynamic parameters estimated in NHPs (Table 1). Yellow: 400 mg bid; orange: 600 mg bid; copper: 600 mg tid.

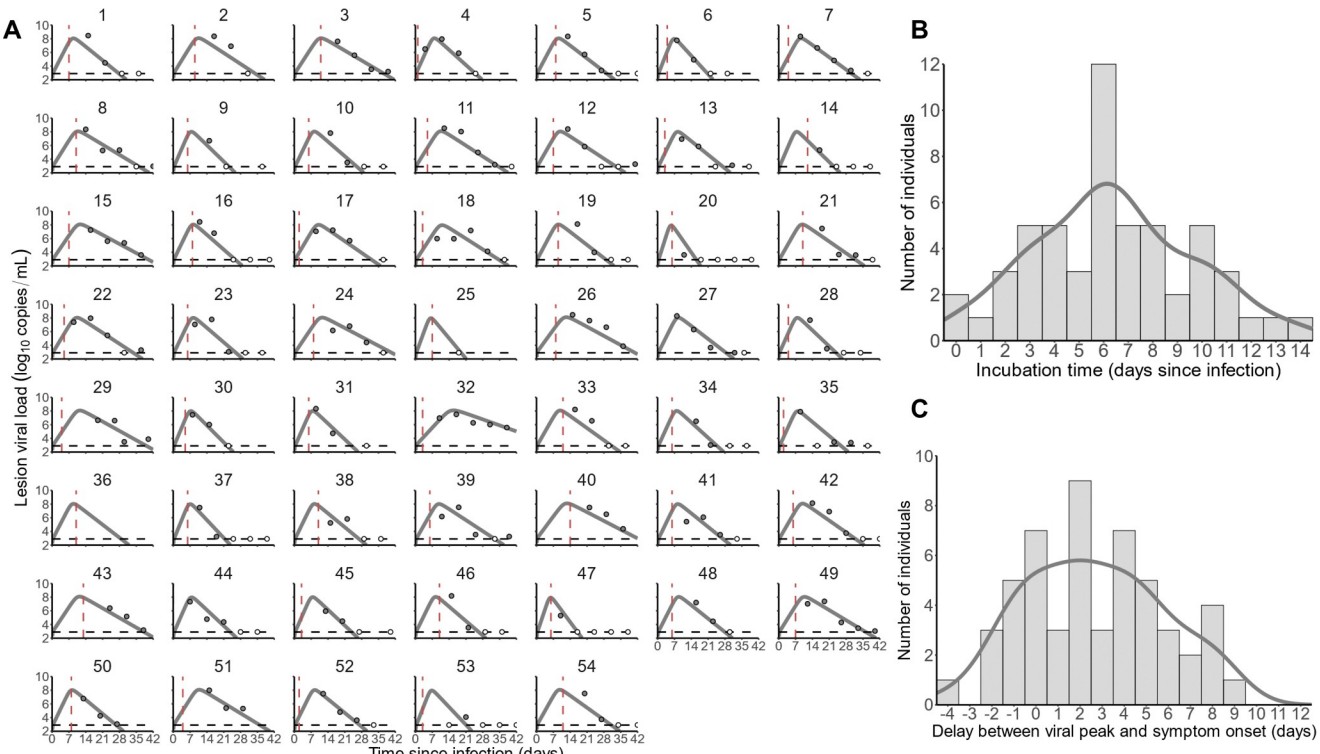

**Fig 3. Mpox viral dynamics in skin lesions of non-antivirally treated individuals ($N = 54$).** (**A**) Viral load in skin lesions (circles) and model predictions (grey). Empty circles indicate data below the limit of quantification by qPCR (LOQ); red dashed lines indicate symptom onset, and horizontal dashed lines indicate the LOQ. (**B**) Distribution of the incubation period. (**C**) Distribution of the delay between predicted peak viral load and symptom onset.

values, only $p \times \mu \times T_0$ was estimable ($1.32 \times 10^6$ copies/mL/day), with sensitivity analysis being performed (S7 Fig). The model estimated a median viral peak equal to 8.0 $\log_{10}$ copies/mL (IQR = 8.0 to 8.0), occurring after a median delay of 9 days after infection (IQR = 8 to 11), i.e.,

**Table 1. Estimates from humans and NHPs used to predict tecovirimat effect in mpox-infected humans.**

| Parameter | Estimate (RSE%) |
|---|---|
| **Parameter estimates from human viral kinetic model ($N = 54$)** | |
| $R_0$ | 5.60 (21%) |
| $p\mu T_0$ (copies/mL/day) | $1.32 \times 10^6$ (72%) |
| $\delta$ (cells/day) | 0.64 (10%) |
| $\omega_{R0}$ | 0.22 (57%) |
| $\omega_p$ | 0.57 (133%) |
| $\omega_\delta$ | 0.30 (16%) |
| $\sigma_{add}$ ($\log_{10}$ copies/mL) | 1.30 (7%) |
| **Parameter estimates from NHP model ($N = 24$)** | |
| $EC_{50}$ (ng/mL) | 1.6 (44%) |
| $k_{e0}$ (per day) | 0.004 (61%) |

RSE, relative standard error; $R_0$, within-host basic reproduction number; $p\mu T_0$, number of infectious virions produced from infected cells; $\delta$, loss rate of infected cells; $EC_{50}$, tecovirimat concentrations inhibiting 50% of viral production; $k_{e0}$, drug transfer rate between plasma and effect compartments; $\omega_\theta$, interindividual variability on parameter $\theta$; $\sigma_{add}$, additional error on viral load.

about 2 days after symptom onset (Fig 3B and 3C). Our analyses also predicted a median duration of positive viral culture in skin lesions of 10 days (IQR = 9 to 12) and a median time to viral clearance of 28 days after infection (IQR = 25 to 34).

The model also estimated a median within-host basic reproduction number $R_0$ = 5.6. This implies that to be fully effective on viral kinetics, an antiviral treatment would require to be given before peak VL with an antiviral activity larger than 1 to 1/5.6 = 82% (see Materials and methods). In other words, any antiviral agent would need to reach concentrations above its $EC_{90}$ to substantially shorten the time to viral clearance.

**Predicting tecovirimat antiviral effect on viral dynamics in humans.** Finally, we predicted tecovirimat activity on mpox dynamics in patients, using model parameters estimated in NHPs and healthy volunteers, as well as in infected individuals. The analysis considered different 14-day tecovirimat dosing regimens, initiated either as immediate postexposure prophylaxis or upon symptom onset (Figs 4 and S8). Viral metrics prediction was also performed for $EC_{50}$ or Hill coefficient higher than identified in NHPs (S9 and S10 Figs).

Without treatment, mpox peaked in skin lesions in median at 8.0 $\log_{10}$ copies/mL on D9 postinfection, becoming undetectable by qPCR in median on D29, and by viral culture on D16. Oral tecovirimat for 14 days given since symptom onset was predicted to reduce the median time from infection to undetectable virus by qPCR and by viral culture by 6 and 4 days, respectively, with a duration of positive viral culture shortened by 5 days, irrespective of the administered dose. As VL generally peaked shortly after symptom onset, the model predicted a minimal impact on median peak VL with a reduction by only 0.5 to 0.7 $\log_{10}$ copies/mL, and a probability of peak VL being below the infectivity threshold (meaning very low risks of contagiousness) between 8% and 18%, depending on the dose. Postexposure prophylaxis further reduced the median peak VL by 1.2, 1.4, and 1.5 $\log_{10}$ copies/mL, respectively. Additionally, this strategy might reduce the median duration of positive viral culture by 7 to 10 days depending on the dose, while leading to a large proportion of individuals remaining noninfectious at all time, as suggested by a probability of peak VL below the culture threshold, equal to 26%, 40%, and 54% at dosing regimen of 400 mg bid, 600 mg bid, and 600 mg tid, respectively.

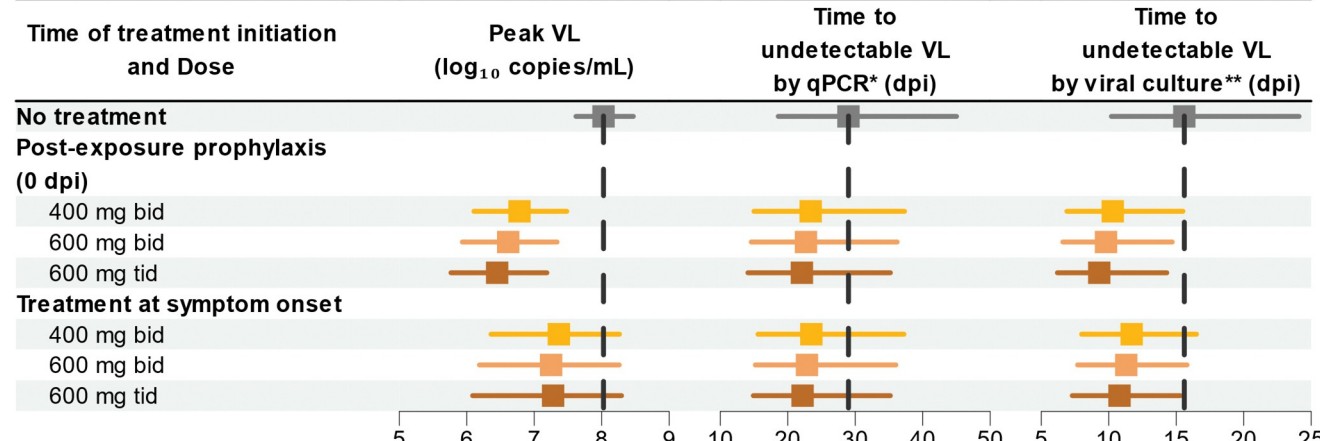

**Fig 4. Predicted effects of tecovirimat on peak viral load (VL), time to undetectable VL by qPCR, and by viral culture, for different dosing regimens and initiation time.** Predictions shown as median and 90% prediction interval, assuming a 14-day treatment course; dpi: days postinfection; *limit of quantification (LOQ) by qPCR: 2.9 $\log_{10}$ copies/mL; **LOQ by viral culture: 6.5 $\log_{10}$ copies/mL.

## Discussion

Using data from infected NHPs and healthy participants treated with tecovirimat, as well as viral dynamics from non-antivirally treated patients, we provided a comprehensive insight in tecovirimat expected antiviral effect in mpox-infected individuals. Tecovirimat could shorten the time to viral clearance and the duration of positive viral culture, provided that treatment is initiated before viral peak (which occurs about 2 days after symptom onset), or even earlier, ideally as an immediate postexposure prophylaxis. This suggests potential benefits in shortening infectious periods following early administration of tecovirimat [7].

Bridging data from NHPs to humans is contingent upon several hypotheses. First, the experimental model of mpox infection in NHPs is more severe than in humans, with an extensive and prolonged viremia in all animals, compared to the transient and low-level viremia generally observed in humans [4,6,7]. Additionally, the NHP model suggested that mpox infection could modify tecovirimat PK, requiring drug exposure analysis in clinical studies [10,14]. Nevertheless, the model allowed us to precisely measure tecovirimat effect on viral replication and, hence, to estimate the concentration–effect relationship of tecovirimat in vivo. Data in untreated individuals originate from the largest prospective analysis of mpox dynamics so far [7]. Viral replication was observed in several compartments, namely, oropharynx, semen, rectum, skin lesions, and blood. Unfortunately, only skin lesions showed sufficiently high viral replication to build a mathematical model and estimate viral dynamic parameters; therefore, our approach could not be used to predict dynamics in other compartments. As the infection date was known, we could reconstruct viral kinetic trajectories from infection to cohort inclusion, estimate the basic within-host reproduction number, $R_0$, and obtain a PD target to effectively prevent viral replication.

The target cell-limited model used to characterize viral dynamics is well established and has been employed in many other acute viral infections, e.g., influenza, Zika, and SARS-CoV-2 [15–18]. This model also assumes that viral clearance is mediated by target cell limitation, which is a very strong assumption. This might be incorrect for human mpox lesions as most of genital skin is left uninfected, even during severe infections [4,19]. More complex models could also be relevant, in particular to understand how the immune response develops, in connection with symptom onset and/or infection resolution. As we did not have access to immunological data, we here made the conservative assumption that the immune response was constant over time, reflected by a loss rate of infected cells, noted $\delta$. This may lead to underestimate the effect of early treatment and how it may synergize with innate immunity [17]. Also, the model was not used to identify tecovirimat effects in other locations and might oversimplify how spatial constraint and innate immune effects impact virus spread. However, data suggest that tecovirimat is well distributed in reproductive compartments, indicating that it could also reduce the risk of sexual transmission [20]. Lastly, the model focused on viral dynamics, but how this may translate in terms of symptoms and clinical evolution will need to be investigated.

Another important assumption is that the concentration–effect relationship in human skin lesions was predicted using data obtained with another mpox strain, Zaire 79, and blood viral kinetics. Recently, an in vitro $EC_{50}$ of 4.8 ng/mL has been identified for the 2022 strain [21], which is close to the value of 1.6 ng/mL (90% PI = 1.5 to 13.4 ng/mL) estimated with our model. Lacking data on tecovirimat in skin lesions, we hypothesized that drug concentrations evolved over time similarly in both infected humans and NHPs in the effect compartment with rapid accumulation in the effect compartment and a long half-life after treatment cessation. This will require further investigation by analyzing tissue distribution of tecovirimat in different locations, including skin, and how this may guide optimal treatment duration. In

mice, tecovirimat was detected in all tissues 24 hours postdose and remained detectable until 168 hours postdose [22]. In humans, at steady state, tecovirimat concentrations in semen may exceed 50 ng/mL, leading to sustained mpox clearance from semen samples after the first week of treatment [20]. Nevertheless, the $EC_{50}$ was reported based on an effect compartment, lacking pharmacological data to verify this part of the model. Consequently, it might be more clinically beneficial to propose plasma concentrations as a PK/PD target. To evaluate whether plasma concentrations are associated with antiviral efficacy in the effect compartment, using our model, we analyzed simulated plasma concentrations by groups of adequate antiviral efficacy and those remaining below the critical threshold. However, our model could not identify any differences in plasma drug concentrations between these groups (not shown). This prediction will need revisiting with upcoming PK/PD clinical results.

In conclusion, our analysis shows that tecovirimat at recommended dosing regimens could shorten the time to viral clearance and reduce the duration of positive viral culture.

## Materials and methods

### Ethics statement

The NHP study was conducted in compliance with US FDA 21 CFR Part 58 (Good Laboratory Practice for Nonclinical Laboratory Studies). Exceptions to US FDA 21 CFR Part 58 include the manufacturing and analysis of the test article, which will be performed under GMP, and characterization of the test pathogen (MPXV). The protocol also complied with all applicable sections of the Final Rules of the Animal Welfare Act regulations (9 CFR Parts 1, 2, and 3) and Guide for the Care and Use of Laboratory Animals—National Academy Press, Washington, DC 1996 (the Guide). This study was conducted in LBERI's (Lovelace Biomedical and Environmental Research Institute, Albuquerque, New Mexico) AAALAC (Association for the Assessment and Accreditation of Laboratory Animal Care)-accredited facility. The study was supported by the Division of Microbiology and Infectious Diseases, National Institute of Allergy and Infectious Diseases, National Institutes of Health (contract HHSN266200600014C), and the Biomedical Advanced Research and Development Authority (contract HHSO100201100001C).

The study protocol of human viral dynamics [7] was approved by the Ethics Committee of the Hospital Germans Trias i Pujol (PI-22-156), and written informed consent was obtained from all participants before enrolment. All investigation were conducted according to the principles expressed in the Declaration of Helsinki.

### Investigating tecovirimat concentration-effect relationship in NHPs

**Study design.**   The experimental procedure and settings were described previously [23]. In brief, 24 macaques (*Macaca fascicularis*) were included with age 2 to 4 years and weight 2.8 to 4.0 kg. On D0, all monkeys were infected with a lethal intravenous (IV) dose of $5 \times 10^7$ pfu of mpox Zaire 79 strain. All animals were randomized into 4 groups of 3 males and 3 females each that were administered vehicle or oral tecovirimat at 3, 10, or 20 mg/kg/day for 14 days (Fig 1A), starting from D4.

**Blood sampling.**   Blood was collected on D0 (immediately following IV inoculation), D3, D6, D9, D12, D15, D18, D21, D24, and D28 postinfection (S1 Fig). Note that in this model of lethal mpox infection, all untreated animals had to be euthanized between D12 and D16. Plasma concentrations of tecovirimat were collected prior to dosing and at 0.5, 1, 2, 3, 4, 6, 8, 12, and 24 hours postdosing on D4, D10, and D17, i.e., on Days 1, 7, and 14 of treatment.

**Modeling viral dynamics in absence of treatment.**   A target cell-limited model was used to fit mpox viral dynamics (Fig 1B) [17,18,24]. The model includes 3 types of cell populations:

target cells ($T$), infected cells in an eclipse phase ($I_1$), and productively infected cells ($I_2$). Target cells are infected at a constant infection rate $\beta$ (mL/virion/day). After a mean eclipse phase of $1/k$ days, infected cells ($I_1$) become productively infected cells ($I_2$), producing virions at rate $p$, and are lost at a per capita rate $\delta$. The virions generated can be infectious ($V_I$) with proportion $\mu$, while the $(1-\mu)$ remaining proportion is noninfectious ($V_{NI}$). The total VL, measured in DNA copies, is the sum of infectious and noninfectious viral particles, both cleared at the same rate ($c$, for virions produced de novo). As inoculation was performed with a high IV dose, we added to the NHP model a compartment for the inoculum (Fig 1B) that distinguishes the injected virus ($V_s$) from the virus produced de novo ($V_I$ and $V_{NI}$) [25]. Uninfected target cells ($T$) can be infected ($I_1$) either by infectious viruses ($V_I$) or inoculum ($V_s$). Virions from the inoculum are cleared at rate $c_i$. Accordingly, the model can be written as follows (Equations 1):

$$dT/dt = -\beta V_I T - \mu \beta V_s T$$

$$dI_1/dt = \beta V_I T + \mu \beta V_s T - kI_1$$

$$dI_2/dt = kI_1 - \delta I_2$$

$$dV_I/dt = \mu p I_2 - cV_I$$

$$dV_{NI}/dt = (1 - \mu)p I_2 - cV_{NI}$$

$$dV_s/dt = -c_I V_s - \mu \beta V_s T$$

The basic within-host reproduction number $R_0$, defined by the number of secondary infected cells resulting from one infected cell in a population of fully susceptible cells, $T_0$, is defined by $R_0 = \beta p T_0 \mu / c\delta$.

**Tecovirimat pharmacokinetics in NHPs.** Tecovirimat individual concentrations in NHP plasma, $C_p(t)$, were predicted using a previously developed PK model assuming a first-order absorption, with weight- and dose-dependent parameters (S2 Table) [14].

**Predicting tecovirimat concentration–effect relationship.** Based on its mechanism of action, tecovirimat is assumed to inhibit viral production $p$, and linked, not to plasma, but intracellular concentrations [9]. Thus, the PK/PD relationship is characterized by a transfer constant $k_{e0}$ between central and effect compartments $dC_e/dt = k_{e0}(C_p - C_e)$, and the therapeutic effect, $\epsilon(t) = C_e^h(t)/(C_e^h(t) + EC_{50}^h)$, where $C_e(t)$ is the effect concentrations at instant $t$; $h$, the Hill coefficient, and $EC_{50}$, the drug concentration producing 50% of the maximal effect. Hence, differential equations for $V_I$ and $V_{NI}$ are modified as follows (Equations 2):

$$dV_I/dt = \mu p[1 - \epsilon(t)]I_2 - cV_I$$

$$dV_{NI}/dt = (1 - \mu)p[1 - \epsilon(t)]I_2 - cV_{NI}$$

**Assumptions on parameter values.** To ensure parameter identifiability of the viral dynamic model, we fixed several parameters of the viral dynamic model. The clearance rates were fixed to 10 and 20 virions/day for virions produced de novo ($c$) and from the inoculum ($c_i$), respectively [25]. As only the product $p \times T_0$ is identifiable, we fixed the density of target cells, $T_0$, to $10^5$ cells/mL for a typical macaque plasma volume of 300 mL [17]. Note that the value of $T_0$ was chosen to be plausible and does not affect our main findings regarding the basic reproduction number $R_0$, the half-life of infected cells, and the antiviral effect [17]. The

number of nonproductively and productively infected cells at the infection time are set to 0 [$I_1(0) = I_2(0) = 0$]. Constant $k$ was fixed to 4 per day (representing a mean eclipse phase of 6 hours). To evaluate our prediction reliability, sensitivity analysis was performed by varying fixed parameters $c$ and $c_i$ between [3–40] virions/day. The Hill coefficient, $h$, was fixed to 1, followed by a sensitivity analysis on virological predictions with higher fixed values between [2–5] (S4 Fig).

**Model fitting strategy.** We fitted the PK and viral dynamic models to data from all animals simultaneously taking a nonlinear mixed-effect modelling approach. All estimations were performed by computing the maximum likelihood estimator using the stochastic approximation expectation–maximization (SAEM) algorithm implemented in Monolix Software. Random effects with SD below 0.1 or associated with RSE >100% were removed using a backward procedure and kept when resulting in BIC increased by no more than 2 points. Goodness of fit was assessed by visually inspecting individual fits and residual scatter plots.

**Model predictions.** The final model was used to extrapolate antiviral efficacy of tecovirimat in NHPs at different dosing regimens at doses of 3, 10, or 20 mg/kg/day administered for 14 days, and this was compared to viral dynamics predicted in absence of treatment. Prediction intervals were obtained by sampling 1,000 simulated individuals in the estimated distribution of the population parameters.

## Predicting PK/PD tecovirimat in humans

**Predicting concentration–effect relationship of tecovirimat in humans.** We used a pharmacokinetic model previously developed in healthy volunteers receiving different doses of tecovirimat to predict the plasma drug concentration of tecovirimat (S2 Table). Using this model and the parameters characterizing the concentration/effect relationship identified in NHPs (i.e., $k_{e0}$ and $EC_{50}$), we predicted tecovirimat antiviral efficacy over time under different dosing regimens in humans. A sensitivity analysis on predicted viral metrics, in which $EC_{50}$ or Hill coefficient can be higher in humans, was also performed to ensure the conservative approximation of in vivo efficacy (S9 and S10 Figs). Because all infected participants in our study were men (see below), the simulation focused on men. We used a similar weight of 78.4 kg for all individuals, which is the median value observed in healthy volunteers (see Results). Prediction intervals were obtained from 1,000 simulated individuals.

As an exploratory analysis, we also tested if weight could modify these predictions, relying on the dose adjustment guidelines of tecovirimat [14].

## Modeling lesion viral dynamics in untreated patients

**Study design and population.** The study population was previously presented in detail [7]. In brief, this observational prospective multicenter cohort included adults over 18 years without severe mpox infection (defined as requiring hospitalization), having symptom onset within the previous 10 days. In order to reconstruct viral dynamics, we only here analyzed the 54 individuals (all male) with known infection date (defined by suspected sexual intercourse with mpox-infected partner) and only focus on viral dynamics in skin lesions. VL in other locations (oropharynx, semen, rectum, and blood) were too low and transient to reliably estimate viral kinetic parameters (see Discussion).

**Sample collection.** Participants were required taking self-collected samples on Days 1, 8, 15, 22, 29, and 57 after the screening visit. All samples were analyzed to detect mpox DNA by qPCR, with a lower limit of quantification of 2.9 $\log_{10}$ copies/mL [7]. None of the samples with VL below 6.5 $\log_{10}$ copies/mL had a positive viral culture, while the probability of positive

culture was equal to 70% in samples with VL above 6.5 $\log_{10}$ copies/mL, which is proposed as the threshold for infectivity [7].

**Model building and parameter estimation.** We used a similar structure for the viral dynamic model to fit to lesion viral dynamics as in NHPs. Because only the product of $p \times T_0$ is identifiable, we fixed $T_0$ by assuming that a skin lesion of 1.0 cm in diameter has an average surface area of 50 mm$^2$ [26,27], and the density of human keratinocytes is 47,000 cells/mm$^2$ [28], which give an amount of $2.3 \times 10^6$ initial target cells. As only a fraction of these cells is actually susceptible to infection, we set this fraction to 10%, thus $T_0$ to $2.3 \times 10^5$ cells. The volume would be approximately 523 mm$^3$, or 0.5 mL, assuming that a skin lesion is spherical. Finally, the proportion of infectious virus $\mu$ was fixed to $10^{-3}$, the median of values $[10^{-4}$–$10^{-2}]$ that were reported in the literature for other viral infections [15–18]. Sensitivity analyses were performed to evaluate the impact of those assumptions, with fraction of sensitive target cells of 1% and 100%, distribution volume of 0.125 and 0.25 mL, and proportion of infectious virus $\mu$ of $10^{-4}$ and $10^{-2}$.

### Projecting the antiviral effect of tecovirimat on viral dynamics

Finally, we combined the PK and the viral dynamic models to predict VL in patients receiving tecovirimat. We considered that treatment initiation could either start immediately after infection (t = 0) or after symptom onset (using the same distribution for the incubation period as found in the data), and different doses of 400 mg bid, 600 mg bid, or 600 mg tid administered for 14 days.

To assess the impact of these different treatment strategies on viral dynamics, 1,000 individuals of 78.4 kg were sampled from the estimated distribution of the PK and viral dynamic parameters, and we calculated the following metrics in each individual: peak VL, time to undetectable virus by PCR (VL <2.9 $\log_{10}$ copies/mL), time to undetectable virus by culture (VL <6.5 $\log_{10}$ copies/mL), and duration of positive viral culture (duration of VL >6.5 $\log_{10}$ copies/mL) [7].

### Supporting information

**S1 Data.** Data for Figs 1–4, S1, S3, S5, S6 and S8–S10. **Data A in S1 Data** (Data description for Fig 1). dosekg: dose (mg) received per kg body weight; time: time postinfection (days); eps, eps05, eps95: median, 5th, and 95th percentile, respectively, for antiviral effect (numeric between 0 and 1); lvl, lvl05, lvl95: median, 5th, and 95th percentile, respectively, for blood viral load ($\log_{10}$ copies/mL). **Data B in S1 Data** (Data description for Fig 2). dose: dose received (mg); time: time postinfection (days); cc, cc05, cc95: median, 5th, and 95th percentile, respectively, for tecovirimat plasma concentrations (ng/mL); eps, eps05, eps95: median, 5th, and 95th percentile, respectively, for antiviral effect (numeric between 0 and 1). **Data C in S1 Data** (Data description for Fig 3). ID: subject identity; time: time postinfection (days); incub: time of symptom onset (days postinfection); lvl: lesion viral load ($\log_{10}$ copies/mL); cens: 0 for observed value above the limit of detection (2.9 $\log_{10}$ copies/mL), 1 otherwise; TTP: time to viral peak (days postinfection); diff: delay between viral peak and symptom onset (days). **Data D in S1 Data** (Data description for Fig 4). initday: time of treatment initiation; dose: dose received; maxvl05, maxvl50, maxvl95: 5th, 50th, and 95th percentile, respectively, for peak viral load ($\log_{10}$ copies/mL); TTP05, TTP50, TTP95: 5th, 50th, and 95th percentile, respectively, for time to viral peak (days postinfection); lloq2.9q05, lloq2.9q50, lloq2.9q95: 5th, 50th, and 95th percentile, respectively, for time to undetectable virus by qPCR (days postinfection); lloq6.5q05, lloq6.5q50, lloq6.5q95: 5th, 50th, and 95th percentile, respectively, for time to undetectable virus by viral culture (days postinfection); dur6.5q05, dur6.5q50, dur6.5q95: 5th,

50th, and 95th percentile, respectively, for duration of positive viral culture (days). **Data E in S1 Data** (Data description for S1 Fig). ID: animal identity; time: time postinfection (days); lvl: blood viral load ($\log_{10}$ copies/mL); cens: 0 for observed value above the limit of detection, 1 otherwise; dosekg: dose (mg) received per kg body weight; eps: predicted antiviral effect (numeric between 0 and 1). **Data F in S1 Data** (Data description for S3 Fig). ID: animal identity; pkday: dosing day; timedose: time since last dose (hour); conc: tecovirimat plasma concentration (ng/mL); type: fit for model prediction, obs for observed concentration; dosekg: dose (mg) received per kg body weight. **Data G in S1 Data** (Data description for S5 Fig). weight: body weight (kg); time: time postinfection (days); cc, cc05, cc95: 5th, 50th, and 95th percentile, respectively, for tecovirimat plasma concentrations (ng/mL); eps, eps05, eps95: 5th, 50th, and 95th percentile, respectively, for antiviral effect (numeric between 0 and 1). **Data H in S1 Data** (Data description for S6 Fig). ID: subject identity; time: time postinfection; lvl: lesion viral load ($\log_{10}$ copies/mL); cens: 1 if simulated value below the limit of detection (2.9 $\log_{10}$ copies/mL), 0 if observed data; empirical_median, empirical_lower, empirical_upper: 50th, 5th, and 95th percentile, respectively, for empirical data ($\log_{10}$ copies/mL); theoretical_median_median, theoretical_median_piLower, theoretical_median_piUpper: 50th, 5th, and 95th percentile, respectively, for predicted median ($\log_{10}$ copies/mL); theoretical_lower_median, theoretical_lower_piLower, theoretical_lower_piUpper: 50th, 5th, and 95th percentile, respectively, for predicted 5th percentile ($\log_{10}$ copies/mL); theoretical_upper_median, theoretical_upper_piLower, theoretical_upper_piUpper: 50th, 5th, and 95th percentile, respectively, for predicted 95th percentile ($\log_{10}$ copies/mL). **Data I in S1 Data** (Data description for S8 Fig). time: time postinfection; dose: dose received; rate2.9, rate2.9q05, rate2.9q95: 50th, 5th, and 95th percentile, respectively, for the percentage of detectable virus by qPCR; rate6.5, rate6.5q05, rate6.5q95: 50th, 5th, and 95th percentile, respectively, for the percentage of detectable virus by viral culture; **Data J in S1 Data** (Data description for S9 Fig). dose: dose received; maxvl05, maxvl50, maxvl95: 5th, 50th, and 95th percentile, respectively, for peak viral load ($\log_{10}$ copies/mL); TTP05, TTP50, TTP95: 5th, 50th, and 95th percentile, respectively, for time to viral peak (days postinfection); lloq2.9q05, lloq2.9q50, lloq2.9q95: 5th, 50th, and 95th percentile, respectively, for time to undetectable virus by qPCR (days postinfection); lloq6.5q05, lloq6.5q50, lloq6.5q95: 5th, 50th, and 95th percentile, respectively, for time to undetectable virus by viral culture (days postinfection); dur6.5q05, dur6.5q50, dur6.5q95: 5th, 50th, and 95th percentile, respectively, for duration of positive viral culture (days). EC50x: $EC_{50}$ fixed value for simulation compared to the final estimate (times). **Data K in S1 Data** (Data description for S10 Fig). dose: dose received; maxvl05, maxvl50, maxvl95: 5th, 50th, and 95th percentile, respectively, for peak viral load ($\log_{10}$ copies/mL); TTP05, TTP50, TTP95: 5th, 50th, and 95th percentile, respectively, for time to viral peak (days postinfection); lloq2.9q05, lloq2.9q50, lloq2.9q95: 5th, 50th, and 95th percentile, respectively, for time to undetectable virus by qPCR (days postinfection); lloq6.5q05, lloq6.5q50, lloq6.5q95: 5th, 50th, and 95th percentile, respectively, for time to undetectable virus by viral culture (days postinfection); dur6.5q05, dur6.5q50, dur6.5q95: 5th, 50th, and 95th percentile, respectively, for duration of positive viral culture (days). h: $h$ fixed value for simulation. (ZIP)

**S1 Fig. Individual fits for blood viral load from tecovirimat PK/PD model in NHPs.** Blood viral load (circles), model predictions for blood viral load (bold solid curves), and for antiviral effect (transparent solid curves). Empty circles indicate data below the limit of quantification by qPCR (LOQ), and horizontal dashed lines indicate the LOQ. Grey: no treatment; green: 3 mg/kg/day; olive: 10 mg/kg/day; orange: 20 mg/kg/day.
(TIF)

**S2 Fig. Sensitivity analysis on fixed parameters in NHP PK/PD model.** Estimation was performed with different value for each fixed parameter using the final model. Estimated population median values reported for $EC_{50}$ and $k_{e0}$. Different colors represent estimates with relative standard error below 50% (white), 50%-100% (light orange), 100%-200% (orange), and over 200% (dark orange). The reference model is $c = 10$ virions/day and $c_I = 20$ virions/day (black border).
(TIF)

**S3 Fig. Individual fits for plasma concentrations from tecovirimat PK/PD model in NHPs.** (**A**) Day 4 postinfection (Day 1 of treatment); (**B**) Day 10 postinfection (Day 7 of treatment); (**C**) Day 17 postinfection (Day 14 of treatment). Observed concentrations presented by circles. Model predictions displayed by solid curves. Green: 3 mg/kg/day; olive: 10 mg/kg/day; orange: 20 mg/kg/day.
(TIF)

**S4 Fig. Sensitivity analysis on NHP model estimates with different *h* fixed values.** Estimation was performed with different value for each *h* value. Different colors represent estimates with relative standard error below 50% (white), 50%-100% (light orange), 100%-200% (orange), and over 200% (dark orange). $\mu$, proportion of infectious virions; $R_0$, within-host basic reproduction number; $pT_0$, number of virions produced from infected cells; $\delta$, loss rate of infected cells; $EC_{50}$, tecovirimat concentrations inhibiting 50% of viral production; $h$, Hill coefficient in the equation of drug concentration–effect relationship; $k_{e0}$, drug transfer rate between plasma and effect compartments; $\omega_\theta$, interindividual variability on parameter $\theta$; $\sigma_{add}$, additional error on viral load. The reference model is $h = 1$ (black border).
(TIF)

**S5 Fig. Projecting tecovirimat PK and PD in mpox-infected humans receiving weight-adjusted doses.** (**A**) Tecovirimat plasma concentrations over time. (**B**) Tecovirimat antiviral effect over time. Predictions shown as median and 90% prediction interval, assuming a 14-day treatment course, using pharmacokinetic (PK) parameters estimated in healthy volunteers (S2 Table), and pharmacodynamic (PD) parameters in NHPs (S1 Table). Golden yellow: 200 mg bid for 20 kg; yellow: 400 mg bid for 40 kg; sandy brown: 600 mg bid for 60 kg; dark orange: 600 mg bid for 80 kg; copper: 600 mg bid for 100 kg; brick: 600 mg tid for 120 kg.
(TIF)

**S6 Fig. Visual predictive check plot for viral loads in skin lesions predicted by the human viral kinetic model (*N* = 54).** 5th, 50th, and 95th empirical percentiles are displayed in grey lines, and 90% prediction intervals of predicted percentiles in grey ribbons. Observed data are presented by circles, including data below the limit of quantification denoted by white circles.
(TIF)

**S7 Fig. Sensitivity analysis on fixed parameters in the human viral kinetics model.** (**A**) Distribution volume fixed to 0.5 mL. (**B**) Distribution volume fixed to 0.25 mL. (**C**) Distribution volume fixed to 0.125 mL. Estimation was performed with different value for each fixed parameter using the final model. Estimated population median values reported for $R_0$ and $\delta$. White and light orange colors represent estimates with relative standard error below 50% and over 50%, respectively. The reference model is $\mu = 10^{-3}$, $T_0 = 200{,}000$ cells, and the distribution volume of 0.5 mL (black border).
(TIF)

**S8 Fig. Probability of detectable virus over time in human skin lesions in patients receiving treatment at infection time (left), or upon symptom onset (right).** Probability over time of

detectable virus by qPCR (**A**, **B**), or by viral culture (**C**, **D**). Predictions are shown as median and 90% prediction interval, assuming a 14-day treatment course. Limits of quantification (LOQ) by qPCR: 2.9 $\log_{10}$ copies/mL; LOQ by viral culture: 6.5 $\log_{10}$ copies/mL. Grey: no treatment; yellow: 400 mg bid; orange: 600 mg bid; copper: 600 mg tid; dashed lines: time to achieve 90% probability of undetectable virus.
(TIF)

**S9 Fig. Sensitivity analysis on viral metrics in mpox-infected humans with different $EC_{50}$ values.** Lesion viral load simulated over time following different postexposure prophylactic doses for each $EC_{50}$ value. Predictions shown as median and 90% prediction interval, assuming a 14-day treatment course for a 78.4-kg patient, using pharmacokinetic parameters estimated in healthy volunteers (S2 Table), and pharmacodynamic parameters estimated in NHPs (S1 Table); dpi: days postinfection; limit of quantification (LOQ) by qPCR: 2.9 $\log_{10}$ copies/ mL; LOQ by viral culture: 6.5 $\log_{10}$ copies/mL.
(TIF)

**S10 Fig. Sensitivity analysis on viral metrics in mpox-infected humans with different $h$ values.** Lesion viral load simulated over time following different postexposure prophylactic doses for each $h$ value. Predictions shown as median and 90% prediction interval, assuming a 14-day treatment course for a 78.4-kg patient, using pharmacokinetic parameters estimated in healthy volunteers (S2 Table), and pharmacodynamic parameters estimated in NHPs (S1 Table); dpi: days postinfection; limit of quantification (LOQ) by qPCR: 2.9 $\log_{10}$ copies/mL; LOQ by viral culture: 6.5 $\log_{10}$ copies/mL.
(TIF)

**S1 Table. Final estimates for tecovirimat PK/PD model in NHPs ($N = 24$).** PK/PD, pharmacokinetic/pharmacodynamic; NHP, nonhuman primate; RSE, relative standard error; $\mu$, proportion of infectious virions; $pT_0$, number of virions produced from infected cells; $\delta$, loss rate of infected cells; $EC_{50}$, tecovirimat concentrations inhibiting 50% of viral production; $k_{e0}$, drug transfer rate between plasma and effect compartments; $\omega_\theta$, interindividual variability on parameter $\theta$; $\sigma_{add}$, additional error on viral load.
(DOCX)

**S2 Table. Final estimates for tecovirimat PK models in mpox-infected NHPs and uninfected humans used in our analysis.** PK, pharmacokinetic; NHP, nonhuman primates; wt: weight; dose: dose level in mg/kg; IIV, interindividual variability; IOV, intra-occasion variability; NA, not available.
(DOCX)

**S3 Table. Characteristics of patients included in the analyses.** Data shown as n (%) or median (IQR); *No participants had a CD4 cell count lower than 100 cells/μL; **Highest count of lesions during the follow-up period.
(DOCX)

**S1 Text. The Movie Group.**
(DOCX)

# Acknowledgments

We thank Mathieu Hubert, Anne-Geneviève Marcelin, Romain Palich, and Olivier Schwartz for their critical reading of an earlier version of the manuscript.

## Author Contributions

**Conceptualization:** France Mentré, Jérémie Guedj.

**Data curation:** Bach Tran Nguyen, Clara Suñer, Douglas W. Grosenbach, Jérémie Guedj.

**Formal analysis:** Bach Tran Nguyen, Aurélien Marc.

**Funding acquisition:** France Mentré, Jérémie Guedj.

**Investigation:** Clara Suñer, Michael Marks, Maria Ubals, Águeda Hernández-Rodríguez, María Ángeles Melendez, Dennis E. Hruby, Andrew T. Russo, Oriol Mitjà, Douglas W. Grosenbach.

**Methodology:** France Mentré, Jérémie Guedj.

**Project administration:** France Mentré, Jérémie Guedj.

**Resources:** Clara Suñer, Michael Marks, Maria Ubals, Águeda Hernández-Rodríguez, María Ángeles Melendez, Dennis E. Hruby, Andrew T. Russo, Oriol Mitjà, Douglas W. Grosenbach.

**Software:** Bach Tran Nguyen, Aurélien Marc.

**Supervision:** Jérémie Guedj.

**Visualization:** Bach Tran Nguyen.

**Writing – original draft:** Bach Tran Nguyen, Aurélien Marc, Jérémie Guedj.

**Writing – review & editing:** Bach Tran Nguyen, Aurélien Marc, Clara Suñer, Michael Marks, Maria Ubals, Águeda Hernández-Rodríguez, María Ángeles Melendez, Dennis E. Hruby, Andrew T. Russo, France Mentré, Oriol Mitjà, Douglas W. Grosenbach, Jérémie Guedj.

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
