## [Editor Report · Decision Letter 0]

13 Jul 2023

Dear Dr. Nguyen, 

Thank you for submitting your manuscript entitled "Antiviral efficacy of tecovirimat against mpox: a translational and modeling study" for consideration as a Research Article by PLOS Biology.

Your manuscript has now been evaluated by the PLOS Biology editorial staff, as well as by an academic editor with relevant expertise, and I am writing to let you know that we would like to send your submission out for external peer review.

Once your full submission is complete, your paper will undergo a series of checks in preparation for peer review. After your manuscript has passed the checks it will be sent out for review. To provide the metadata for your submission, please Login to Editorial Manager (https://www.editorialmanager.com/pbiology) within two working days, i.e. by Jul 15 2023 11:59PM.

Kind regards,

Paula

---

Senior Editor

PLOS Biology

---

## [Decision Letter · Decision Letter 1]

16 Oct 2023

Dear Dr. Nguyen,

Please allow me to first apologize for the delay in the processing of your manuscript. This delay is caused by my difficulty in recruiting reviewers for your manuscript, and is further compounded by one referee promising an overdue report but failing to deliver after long delay and multiple chases, which forced us to find alternative reviewers late in the process. I am sorry for this unexpected event, and I thank you for your patience while your manuscript "Antiviral efficacy of tecovirimat against mpox: a translational and modeling study" went through peer-review at PLOS Biology. Your manuscript has now been evaluated by the PLOS Biology editors, an Academic Editor with relevant expertise, and by several independent reviewers.

In light of the reviews, which you will find at the end of this email, we are pleased to offer you the opportunity to address the comments from the reviewers in a revision that we anticipate should not take you very long. We will then assess your revised manuscript and your response to the reviewers' comments with our Academic Editor aiming to avoid further rounds of peer-review, although might need to consult with the reviewers, depending on the nature of the revisions.

Please also address the following policy and editorial issues:

1. We suggest a change in the title along these lines: "Early administration of tecovirimat shortens the time to mpox clearance in a model of human infection".

2. Please describe all sources of funding that have supported your work. This information is required for submission and will be published with your article, should it be accepted. A complete funding statement should do the following:

- Include grant numbers and the URLs of any funder's website. Use the full name, not acronyms, of funding institutions, and use initials to identify authors who received the funding.

- Describe the role of any sponsors or funders in the study design, data collection and analysis, decision to publish, or preparation of the manuscript. If the funders had no role in any of the above, include this sentence at the end of your statement: "The funders had no role in study design, data collection and analysis, decision to publish, or preparation of the manuscript.

3. Please provide a blurb which (if accepted) will be included in our weekly and monthly Electronic Table of Contents, sent out to readers of PLOS Biology, and may be used to promote your article in social media. The blurb should be about 30-40 words long and is subject to editorial changes. It should, without exaggeration, entice people to read your manuscript. It should not be redundant with the title and should not contain acronyms or abbreviations.

4. ETHICS STATEMENT:

-- Please include the full name of the IACUC/ethics committee that reviewed and approved the animal care and use protocol/permit/project license. Please also include an approval number.

-- Please include the specific national or international regulations/guidelines to which your animal care and use protocol adhered. Please note that institutional or accreditation organization guidelines (such as AAALAC) do not meet this requirement.

-- Please include information about the form of consent (written/oral) given for research involving human participants. All research involving human participants must have been approved by the authors' Institutional Review Board (IRB) or an equivalent committee, and all clinical investigation must have been conducted according to the principles expressed in the Declaration of Helsinki.

5. DATA POLICY:

A) Supplementary files (e.g., excel). Please ensure that all data files are uploaded as 'Supporting Information' and are invariably referred to (in the manuscript, figure legends, and the Description field when uploading your files) using the following format verbatim: S1 Data, S2 Data, etc. Multiple panels of a single or even several figures can be included as multiple sheets in one excel file that is saved using exactly the following convention: S1_Data.xlsx (using an underscore).

B) Deposition in a publicly available repository. Please also provide the accession code or a reviewer link so that we may view your data before publication.

Regardless of the method selected, please ensure that you provide the individual numerical values that underlie the summary data displayed in the following figure panels as they are essential for readers to assess your analysis and to reproduce it: Figures 1CD, 2AB, 3ABC, 4, and Supplementary Figures S1, S3ABC, S4AB, S5, S7ABCD.

**Please also ensure that figure legends in your manuscript include information on where the underlying data can be found, and ensure your supplemental data file/s has a legend.**

**IMPORTANT - SUBMITTING YOUR REVISION**

*Resubmission Checklist*

*Published Peer Review*

*PLOS Data Policy*

*Blot and Gel Data Policy*

Sincerely,

Paula

---

Senior Editor

PLOS Biology

REVIEWS:

Reviewer #1: Joshua T Schiffer. Viral dynamics, mathematical models.

Reviewer #2: Mathematical modelling of virus-host interactions.

Reviewer #3: Antiviral drug pharmacokinetic-pharmacodynamic relationships.

Reviewer #1: This is a great paper that is clearly written and captures the benefit of mathematical modeling for clinical trial simulation. This is really the perfect use of modeling to inform future experiments and trials. Consideration of viral dynamics in addition to PK/PD models adds another critical dimension toward optimizing the likelihood of therapeutic efficacy in clinical trials. I only have minor points:

1) The authors point out that the the EC50 derived based on reducing viremia in non human primates may not be equivalent to that required for reducing viral load in skin lesions in humans. Estimates for other antivirals have suggested higher concentrations needed in vivo relative to in vitro. A sensitivity analysis in which the human in vivo EC50 is higher would provide a more conservative approximation of in vivo efficacy and would make a nice supplementary figure. It would be interesting to see if doses in humans start to matter more if the actual in vivo EC50 in humans is higher than that in non human primates.

2) The mechanism of action of tecovirimat should be mentioned somewhere in the paper to justify that it impacts viral replication rather than infectivity / viral entry (beta).

3) Was the Hill coefficient considered or estimated in vitro? Could this impact viral load trajectories if varied?

4) This is picky but target cell limitation seems like the wrong term for this infection. Clearly there is not target cell limitation for human Mpox lesions as most genital skin is left uninfected even during severe infections. This is easily visible to the naked eye. While the model implies target cell limitation, I think it would be good to acknowledge that what may really be occurring is a combination of spatial constraint on virus spread as well as innate immune effects. It is fine these details are not captured with the model but the over simplification of the target cell limitation assumption should be noted.

5) Equating culture positivity with infectivity is an oversimplification as no formal surrogate for transmissibility has been established for this virus. I might say that culture is likely to be a more specific marker of infectiousness than PCR.

Reviewer #2: This is an outstanding manuscript that describes a modeling approach to analyze mpox kinetics under tecovirimat treatment in humans. The authors pool data from non human primates, healthy volunteers and a small grou of patients to develop a kinetic model of expected treatment effects in humans.

I can only congratulate the authors on this work. The manuscript addresses a highly relevant topic, and given small patient numbers, the approach taken by the authors is an excellent way to gain information about expected treatment effects and optimize treatment schedules, here leading to the recommendation to use tecovirimat immediately after exposure. The manuscript is clearly written, and the models and methods employed are sound. The results are convincing. I have - and this is an absolute exception - no further requests for changes and find the paper ready for publication in its present form.

Reviewer #3: This is an interesting bridging study to link pharmacokinetic-pharmacodynamic (PK-PD) relationships for tecoviramat in human mpox disease. The manuscript is clearly written and the topic is important for the field. There are some points that would help clarify the methods and findings for readers.

1. There was a difference in macaque PK for mpox infected versus uninfected animals(conc were lower in infected animals). Was this considered for bridging the PK to humans (ie that mpox infected humans could have lower conc compared with healthy volunteers upon which the PK for this study was based?). If not, this should be listed as a limitation.

2. I agree with the basic conclusions from the paper - that the model supports clinically relevant tecoviramat activity for mpox in humans, but it is based on a model that bridges macaque and human data to make estimates. There are many necessary assumptions and the authors address some, but not ones that are unforeseen. I think the conclusions/discussion should acknowledge that unforeseen assumptions could change the model outcomes and the conclusions. Also, the authors should go through the paper and make sure the model findings are referred to as "estimates" or "model predictions" throughout.

3. Please define the omegas and sigma in table 1.

4. Please include a few sentence on key pharmacologic properties of tecoviramat: mechanism of action, plasma protein binding, bioavailability, metabolic fate.

5. The authors mention dose dependent PK but the human models do not include dose on clearance or volume. Was this just in the macaque model?

6. the colors in the figure (yellows and oranges) are difficult to distinguish. consider more contrasting colors, or dashes (etc) to help reader distinguish the different curves/groups.

7. the keo (effect site rate constant) generates a concentration in an effect compartment, but this appears to be completely arbitrary as the authors have no pharmacologic data in such a compartment with which to verify this part of the model. this is relevant for the EC50 which appears to be from the effect site compartment concentration, thus not verifiable. Is it possible to report an EC50 based upon plasma concentrations? This reviewer understands the importance of a lag time to reach the effect compartment concentration, but a plasma EC50 (if available from the model) is much more relevant for the field.

---

## [Editor Report · Decision Letter 2]

17 Nov 2023

Dear Dr Nguyen,

Thank you for the submission of your revised Research Article "Early administration of tecovirimat shortens the time to mpox clearance in a model of human infection" for publication in PLOS Biology. On behalf of my colleagues and the Academic Editor, Ronald Regoes, I am pleased to say that we can in principle accept your manuscript for publication, provided you address any remaining formatting and reporting issues. These will be detailed in an email you should receive within 2-3 business days from our colleagues in the journal operations team; no action is required from you until then. Please note that we will not be able to formally accept your manuscript and schedule it for publication until you have completed any requested changes.

PRESS

Sincerely, 

Paula

---

Senior Editor

PLOS Biology
